# Determining the pneumococcal conjugate vaccine coverage required for indirect protection against vaccine-type pneumococcal carriage in low and middle-income countries: a protocol for a prospective observational study

Jocelyn Chan,[1,2] Cattram D Nguyen,[1,2] Jana Y R Lai,[1] Eileen M Dunne,[1,2]
Ross Andrews,[3,4] Christopher C Blyth,[5,6] Siddhartha Datta,[7] Kim Fox,[8]
Rebecca Ford,[9] Jason Hinds,[10,11] Sophie La Vincente,[1] Deborah Lehmann,[12]
Ruth Lim,[1] Tuya Mungun,[13] Paul N Newton,[14,15] Rattanaphone Phetsouvanh,[14,15]
Willam S Pomat,[9,12] Anonh Xeuatvongsa,[16] Claire von Mollendorf,[1,2]
David A B Dance,[15] Catherine Satzke,[1,17] Kim Muholland,[1,18] Fiona M Russell,[1,19] for
the PneuCAPTIVE Protocol Group

For numbered affiliations see end of article.

**Correspondence to**
Dr Jocelyn Chan;
jocelyn.chan@mcri.edu.au

## ABSTRACT

**Introduction** Pneumococcal conjugate vaccines (PCVs) prevent disease through both direct protection of vaccinated individuals and indirect protection of unvaccinated individuals by reducing nasopharyngeal (NP) carriage and transmission of vaccine-type (VT) pneumococci. While the indirect effects of PCV vaccination are well described, the PCV coverage required to achieve the indirect effects is unknown. We will investigate the relationship between PCV coverage and VT carriage among undervaccinated children using hospital-based NP pneumococcal carriage surveillance at three sites in Asia and the Pacific.

**Methods and analysis** We are recruiting cases, defined as children aged 2–59 months admitted to participating hospitals with acute respiratory infection in Lao People's Democratic Republic, Mongolia and Papua New Guinea. Thirteen-valent PCV status is obtained from written records. NP swabs are collected according to standard methods, screened using *lytA* qPCR and serotyped by microarray. Village-level vaccination coverage, for the resident communities of the recruited cases, is determined using administrative data or community survey. Our analysis will investigate the relationship between VT carriage among undervaccinated cases (indirect effects) and vaccine coverage using generalised estimating equations.

**Ethics and dissemination** Ethical approval has been obtained from the relevant ethics committees at participating sites. The results are intended for publication in open-access peer-reviewed journals and will demonstrate methods suitable for low- and middle-income countries to monitor vaccine impact and inform vaccine policy makers about the PCV coverage required to achieve indirect protection.

### Strengths and limitations of this study

► We describe a novel application of study methods that enable monitoring the indirect effects of pneumococcal conjugate vaccines using pneumococcal carriage and in the absence of baseline data.

► The methods do not measure indirect protection against disease. However, a reduction in vaccine-type (VT) carriage among undervaccinated cases suggests likely reductions in VT disease, since carriage is a precursor for disease The use of molecular serotyping microarray enables detection of multiple serotype carriage and serotype-specific carriage density.

► The inclusion of three sites, which have contrasting vaccine schedules and pneumococcal epidemiology, will enable us to explore factors that may modify the vaccine coverage required to achieve indirect effects; however, variations in methods and settings may impact on the comparability of our results across the three sites.

## INTRODUCTION

Infections due to *Streptococcus pneumoniae* (the pneumococcus), including pneumonia, meningitis and septicaemia, are a leading cause of morbidity and mortality among children and elderly, especially in low- and middle-income countries (LMIC).[1] The main reservoir for the pneumococcus is the human nasopharynx. Pneumococcal carriage peaks in young children, an important group for transmission of pneumococci to older age

groups,[2 3] and is a precursor for disease.[4] There are over 90 known serotypes of pneumococci, with differing capacities for causing disease.[5]

The introduction of the pneumococcal conjugate vaccine (PCV) has resulted in substantial reductions in pneumococcal disease in many settings.[6] These reductions are mediated by direct effects on vaccinated children as well as indirect effects on undervaccinated children and adults through a reduction in transmission and subsequent nasopharyngeal (NP) carriage of pneumococcal serotypes included in the vaccine.[7–10] The indirect effects account for a substantial component of the overall vaccine effect. Following the introduction of the 7-valent PCV (PCV7) into the routine vaccination programme in the USA, twice as many invasive cases were prevented through indirect effects compared with direct effects.[11]

The magnitude of indirect effects following introduction of PCV varies considerably by setting. Reductions in invasive pneumococcal disease (IPD) among adults ranged from 8.8% among adults in Denmark 3 years after PCV implementation to 70% reduction among adults in Taiwan 7 years after PCV implementation.[12] A review of the literature found that higher vaccine coverage, higher baseline rates of pneumococcal disease and greater time elapsed since PCV introduction were all associated with greater degrees of indirect effects.[12] However, the majority of these studies were conducted in high-income country settings using a similar vaccine schedule, with relatively high levels of vaccine coverage.

The threshold of vaccination coverage required to achieve significant indirect PCV effects on either pneumococcal carriage or disease outcomes are not well understood. Studies of NP carriage indicate that there is variability in the coverage required to achieve significant indirect effects. Two studies in the USA examined this question for the 13-valent PCV (PCV13) that superseded PCV7 immunisation. A vaccine coverage of 58% among American Indian children in southwestern USA and 75% coverage among children in Massachusetts resulted in a 50% decline in the prevalence of carriage of six PCV13 serotypes (ie, PCV13 types not included in PCV7) carried by undervaccinated children.[13 14] The USA uses a 3+1 schedule (three primary doses with a booster) and had a catch-up programme up to 5 years of age. Many LMIC use a 3+0 schedule.[15] More data are needed from a range of LMIC to determine the vaccination coverage required to achieve indirect protection from PCVs and to determine factors that may modify the vaccine coverage required for indirect effects, such as baseline carriage prevalence, indicating intensity of transmission, vaccine schedules and use of catch-up campaigns.

Existing systematic reviews indicate that vaccine schedules do not impact on the degree of indirect effects on IPD; however, the studies included in these reviews predominantly use a 3+1 schedule or a 2+1 schedule (two primary doses with a booster) and have limited studies using PCV13.[12 16] Few studies using the 3+0 schedule (three primary doses with no booster) are available

in the published literature, despite the 3+0 schedule being widely used, especially in LMIC. A recent study from Australia, the only high-income country to use the 3+0 schedule, concluded that the booster dose may be important for obtaining indirect protection because of the lower indirect effects observed compared with other high-income country settings.[17]

Despite the gradual introduction of PCVs in LMIC over the last decade, there have been few studies published on the impacts of PCV in these settings.[18 19] This is because the predominant method for assessing vaccine impact, IPD surveillance, is costly and resource intensive to establish—requiring laboratory capacity as well as the collection of large numbers of samples, obtained using aseptic techniques, in order to detect a relatively rare outcome.[20] Furthermore, baseline data prior to vaccine introduction required for impact evaluations are often not available in these settings.

In the absence of IPD surveillance, LMIC require a method to evaluate direct and indirect vaccine effects. We propose using NP carriage surveillance in children hospitalised with acute respiratory infection (ARI) to monitor the indirect effects of PCV13 in the study called PNEUmococcal CArriage in Pneumonia To Investigate Vaccine Effects (PneuCAPTIVE).

In this context, we refer to 'carriage' as the detection of pneumococci in the nasopharynx of a child with ARI. A reduction in vaccine-type (VT) pneumococcal carriage suggests likely reductions in disease due to VT pneumococci, since carriage is a precursor to disease.[4] Existing studies among healthy children have demonstrated reductions in VT carriage following PCV introduction, while overall carriage remains stable due to replacement with non-vaccine type (NVT) carriage.[21 22]

We aim to determine the PCV13 coverage required to demonstrate substantial indirect effects of PCV using NP carriage surveillance in children hospitalised with ARI in three settings within Asia and the Pacific. We are focusing on children under 5 years of age because studies have shown that children are the main reservoir for pneumococci, and reductions in transmission within this age group are likely to result in reductions in VT carriage among older age groups.[2 3 23]

As this is a novel method for determining indirect effects, we will also aim to determine whether changes in patterns of VT and NVT pneumococcal carriage among children with ARI are reflective of changes in serotypes circulating in the community, noting that carriage in our cohort may be more reflective of serotypes causing disease.[5 24]

## OBJECTIVES

Our objectives are to: (1) investigate the relationship between PCV13 coverage and VT carriage among undervaccinated cases, defined as children (indirect effects) aged 2–59 months with an ARI in Lao People's Democratic Republic (Lao PDR), Mongolia and Papua New

Guinea (PNG); (2) describe monthly trends in VT carriage prevalence among cases and contacts. Contacts are defined as children 0–59 months of age, who have slept in the same house as or played with the case during the preceding 3 weeks; (3) investigate the relationship between PCV13 coverage and VT carriage among under-vaccinated contacts (indirect effects) and caregivers living in the community; and (4) compare the PCV13 coverage required to demonstrate indirect effects of PCV13 by site and determine the degree to which site-specific factors, such as pneumococcal carriage rates and densities, vaccine schedule and use of catch-up campaigns, account for differences in the PCV13 coverage required to demonstrate indirect effects.

## METHODS
### Study design
We are conducting prospective hospital-based observational studies in Lao PDR, Mongolia and PNG. We are recruiting children 2–59 months of age presenting with ARI and obtaining NP swabs to determine prevalence and density of VT carriage. We are determining the PCV13 status of each case using written record. Recruitment will occur over at least 3 years and up to 5 years post-PCV13 introduction at each site.

Concurrently, we are determining vaccination coverage at the resident village or subdistrict of each recruited case, using either administrative data or vaccination coverage surveys.

### Study settings
#### Lao PDR site
The Lao PDR PneuCAPTIVE study is embedded within a hospital-based study of ARI aetiology, in collaboration with Lao Oxford Mahosot Hospital-Wellcome Trust Research Unit, the WHO and the Lao PDR Ministry of Health.[25] PCV13 was introduced in October 2013, using a

3+0 schedule at 6, 10 and 14 weeks of age and a catch-up programme up to 12 months of age (table 1). We are recruiting cases at Mahosot Hospital, one of the largest paediatric referral hospitals in Vientiane, the capital of Lao PDR. Recruitment started in December 2013.

#### Mongolia site
The Mongolian PneuCAPTIVE study is embedded within a hospital-based paediatric pneumonia surveillance program to determine vaccine impact, which is conducted in partnership between the Murdoch Children's Research Institute (MCRI), WHO and the Mongolian Ministry of Health. PCV13 was introduced in June 2016, using a modified 2+1 schedule at 2, 4 and 9 months of age, within the two 'Phase 1' districts within Ulaanbaatar, the capital of Mongolia as part of a phased introduction (table 1), with a catch-up programme of two doses 1 month apart for those up to 24 months of age. We are recruiting cases, residing in the two phase 1 districts, at the two district hospitals and the tertiary referral paediatric hospital for Mongolia, the Maternal and Child Hospital. Sampling started November 2015.

#### PNG site
The PNG PneuCAPTIVE study represents a collaboration between the PNG Institute of Medical Research, Telethon Kids Institute, the University of Western Australia and MCRI. It is an extension of a pneumonia aetiology study that commenced in 2013.[26] PCV13 was introduced to PNG in October 2014 using a 3+0 schedule at 1, 2 and 3 months of age (table 1); however, it was not widely distributed in the Eastern Highlands Province until late 2015. We are recruiting cases at the Eastern Highlands Provincial Hospital, the major referral hospital for the Eastern Highland Province, as well as nearby clinics in Goroka, the capital of the Eastern Highlands Province. Recruitment started April 2016. In PNG, we are also recruiting caregivers, as well as contacts, defined as children 0–59 months of age, who have slept in the same house as or played with the case during the preceding 3 weeks. This will enable us to determine whether changes in patterns of VT pneumococcal carriage in the hospitalised cases are reflective of changes within the community, as well as to examine indirect effects in the adult age group.

### Case recruitment and data collection
Participant recruitment and data collection are consistent across the three sites; however, there are some local adaptations to the protocol at each site, which are summarised in table 2. These adaptations are due to the PneuCAPTIVE study being nested within other existing studies, described above.

Cases are eligible for inclusion in the PneuCAPTIVE study if they are 2–59 months of age and presenting with ARI (defined in table 2 below). All cases with fever or respiratory symptoms are screened for inclusion. In Mongolia, we are restricting recruitment to patients living within the two 'Phase 1' districts that have commenced

**Table 1** Key aspects of the 13-valent pneumococcal conjugate vaccination (PCV13) programme by site, Lao People's Democratic Republic (PDR), Mongolia and Papua New Guinea (PNG)

|  | Lao PDR | Mongolia | PNG |
| --- | --- | --- | --- |
| Year of PCV13 introduction | October 2013 | June 2016 | October 2014 |
| Location of PCV13 introduction | National | Two districts in Ulaanbaatar | National |
| PCV13 schedule | 3+0 (6, 10 and 14 weeks) | 2+1 (2, 4 and 9 months) | 3+0 (1, 2 and 3 months) |
| Presence of a catch-up programme | Catch-up of three doses up to 12 months of age | Catch-up of two doses 2 months apart up to 24 months of age | None |

**Table 2** Patient eligibility by site: Lao People's Democratic Republic (PDR), Mongolia and Papua New Guinea (PNG)

| Site | Lao PDR | Mongolia | PNG |
|---|---|---|---|
| Inclusion criteria | 2–59 months of age and presenting with: | | |
| Definition of acute respiratory infection | Fever (parent report or measured) AND one of cough OR dyspnoea OR rhinitis OR abnormal chest auscultation | Cough OR dyspnoea AND tachypnoea* OR hypoxia OR chest indrawing | Cough AND tachypnoea* AND lower chest wall indrawing |
| Exclusion criteria | – | Lives outside phase 1 district. Admitted with pneumonia in the last 14 days. | Lives ≥1 hour outside town OR hospitalisation in past 14 days |
| Recruitment site | Inpatient setting only | Inpatient setting only | Inpatient and outpatient setting |
| Sampling | Monday–Friday | A random sample (33 per month) of all enrolled cases are selected for testing. | Monday–Friday |
| Sampling period | December 2013–November 2019 | November 2015–October 2018† | April 2016–March 2019 |
| Study population | Mahosot Hospital is one of two tertiary-level paediatric hospitals in Vientiane and receives a mix of patients from urban Vientiane city, rural Vientiane province and other provinces. | The two secondary–level district hospitals and a tertiary-level Maternal and Child Health hospital service the vast majority of children in the two districts that received PCV. There are a limited number of paediatric beds at private hospitals in Ulaanbaatar. | The Eastern Highlands Provincial hospital is the sole hospital for the province. Study population includes urban and rural households within 1-hour drive of Goroka. |

*Tachypnoea is defined as ≥50 breaths per minute.
†A 1-year extension (until June 2019) has been sought for the Mongolian site.

PCV13 in 2016. In PNG, we are restricting recruitment to patients living within 1 hour of the town as follow-up is logistically challenging.

Regarding recruitment, in Lao PDR, study staff are screening potential recruits from Monday to Friday each week. However, they are obtaining clinical information from medical records for all eligible cases, including those admitted at weekends to ensure we have a representative sample. In Mongolia, caregivers of all eligible children presenting to the hospital will be approached for recruitment as part of the larger PCV impact study. However, for the purposes of this study, we will select a random sample of 33 cases per month for microbiological testing. In PNG recruitment takes place 4–5 days per week.

After determining eligibility and obtaining informed parental consent, we complete a questionnaire to obtain: demographic data, clinical data, PCV13 status and risk factors for vaccination and NP carriage, including prior antibiotic use (see analysis section below for complete list). Vaccination status is determined using written records, either parent-held immunisation records or health centre administrative records.

We then collect an NP swab according to WHO guidelines and store it in 1 mL skim milk tryptone glucose glycerol (STGG) medium.[27] Swabs are vortexed, aliquoted and stored frozen at −80°C within 8 hours of collection and transported from all three sites to the Pneumococcal Research laboratory at MCRI on dry ice or in liquid nitrogen, where they will be stored at −80°C.

### Laboratory methods

All samples are screened for the presence of pneumococci using real-time quantitative PCR (qPCR) assay targeting the pneumococcal *lytA* gene.[22] Genomic DNA are extracted from 100 µL of STGG using a MagNA Pure LC Machine (Roche) using the DNA Isolation Kit III (Bacteria, Fungi) (Roche) following an enzymatic lysis treatment. The pneumococcal load is estimated by reference to a standard curve.

All swabs that are *lytA* positive or equivocal are molecular serotyped using BµG@S Senti-SP v1.5 microarray (BUGS Bioscience) as previously described.[28]

Serotype-specific pneumococcal density is calculated using the relative abundance of each serotype identified, as determined using microarray and interpreted with the assistance of a Bayesian random effects model as previously described,[28 29] and the overall pneumococcal load as determined by the *lytA* qPCR.

### Key definitions

The primary outcome, VT carriage, is defined as the NP carriage of at least one pneumococcal serotype included in the PCV13 vaccine, that is, serotypes 1, 3, 4, 5, 6A, 6B, 7F, 9V, 14, 18C, 19A, 19F and 23F. In the context of multiple serotype carriage, VT carriage will be defined as the presence of at least one VT serotype regardless of the presence of other serotypes. A secondary outcome is VT carriage density (CFU/mL), which is defined as an aggregate of the serotype-specific density for each of the VTs carried by the case, and will be reported as a continuous

**Table 3** Vaccination coverage data by site, Lao People's Democratic Republic (PDR), Mongolia and Papua New Guinea

| Site | Lao PDR | Mongolia | Papua New Guinea |
|---|---|---|---|
| Source of numerator data | Health centre records | Electronic immunisation record | Community surveys |
| Source of denominator data | Lao PDR Population and Housing Census 2015 | Health centre population register* | |

*All children are required to be registered at the health centre servicing their resident subdistrict in order to receive health services.

variable using a log scale to account for large variations in density and a skewed distribution.

Vaccine history will be defined based on documented evidence of receiving an adequate number of PCV doses to provide a protective immune response against vaccine serotypes at least 14 days prior to study enrolment.[30] For children <12 months of age, 'vaccinated' is defined as two or more PCV13 doses. For children 12 months of age or older, 'vaccinated' is defined as receipt of two doses in the first year of life or at least one dose after the age of 12 months. Conversely, a case will be defined as 'undervaccinated' if they have received less than the adequate number of PCV doses (including those never vaccinated). Sensitivity analyses will be conducted using varying definitions of vaccinated including receiving at least one dose of vaccine at any age.

### Vaccine coverage data collection

For each undervaccinated case recruited (including those with and without VT pneumococci), we are determining their resident village or subdistrict vaccination coverage. In Mongolia and Lao PDR, we are using administrative data. The resident village or subdistrict is identified using relevant administrative codes, which are determined by local staff on enrolment. This determines the health centre's administrative boundary for the provision of immunisation services, whereas in PNG, we are conducting community surveys within 10 days of discharge in the village where the case is living (table 3). We are surveying all children less than 5 years of age, from households within 10–20 min walk of the case, since this is the group of children with whom the case is mostly likely to interact and therefore influence their carriage status.

To determine the reliability of our methods in Lao PDR, we plan on comparing our coverage estimates with a National Immunisation Survey, conducted according to WHO guidelines in 2015, when provincial level estimates from this survey become available. In Mongolia, we have validated the newly introduced electronic immunisation record against clinic health records, finding a high degree of concordance.[31] We are also in the process of validating the population registers at health centres in Mongolia.

### Data management

In Lao PDR and Mongolia, study staff are double-entering data using Research Electronic Data Capture and Microsoft Access (Microsoft Corporation) database, respectively. In PNG, data are checked by a monitor prior to being entered into Filemaker Pro (FileMaker). We are conducting regular double-entry discrepancy checks and logic checks using Stata Statistical Software.

### Analysis

All analyses will be completed using Stata Statistical Software. We will summarise continuous variables using mean and SD (or median and IQR for non-symmetrical data). Categorical variables will be summarised using frequency counts and percentages.

#### Objective 1: relationship between vaccine coverage and indirect effects

We want to investigate the relationship between VT carriage and density among undervaccinated cases (indirect effects) and subdistrict/village PCV coverage. We will use an adaptation of a method used to estimate indirect protection for an oral cholera vaccine, which exploits heterogeneities in vaccine coverage at the subdistrict/village level, comparing VT carriage and density among undervaccinated children from subdistricts/villages with differing levels of vaccine coverage.[32 33]

This will be done using multivariable models with VT carriage or density in ARI cases as the outcome variable and PCV coverage at the child's place of residence at the time of admission as the exposure variable. We will use generalised estimating equations to account for clustering at the subdistrict/village level.

To identify confounders for adjustment, we have constructed a directed acyclic graph (DAG). DAGs include all variables potentially related to exposure and outcome, connected using unidirectional arrows showing causal relationships between variables. The graph identifies potentially confounding pathways and allows investigators to determine variables that should be controlled for to obtain unbiased effect estimates. As there are likely to be unique confounders between sites, we will develop site-specific DAGs. We will use DAGitty.net (V.2.3) software to identify minimally sufficient confounding subsets for adjustment.

For each site, we will construct a similar model using overall pneumococcal carriage as the dependent variable. This model will act as a bias indicator since PCV coverage is not expected to affect levels of overall pneumococcal carriage due to replacement carriage with NVTs, although complete replacement to baseline levels can take several years.[21 34 35] Therefore, we will restrict this analysis to the latter part of the study period, when descriptive analyses indicate the replacement is complete.

To determine whether a higher PCV coverage is required to achieve indirect effects among completely unvaccinated cases compared with undervaccinated cases, we will conduct a sensitivity analysis among children who have never received PCV.

### Objective 2: VT carriage prevalence among cases and contacts by calendar month (PNG only)

Crude and adjusted monthly VT carriage prevalence will be estimated within rolling 7-month intervals to present smooth curves and assess trends over time. This will be done separately for cases and contacts. To account for differences in age between cases and community contacts, the carriage prevalence will be adjusted using direct standardisation (standardised to the case population over the entire study period).

### Objective 3: relationship between vaccine coverage and indirect effects among community contacts (PNG only)

To investigate whether relationship between vaccine coverage and indirect effects as observed among cases with ARI are reflective of the relationship between vaccine coverage and indirect effects in the wider community, we will apply the same model described above to undervaccinated community contacts. We will construct multivariable models with VT carriage and density among undervaccinated community contacts as the outcome variable and PCV coverage at the child's place of residence at the time of admission as the exposure variable. We will be using the same vaccine coverage data as for the cases.

### Objective 4: comparison of vaccine coverage required for indirect effects across sites

We will compare differences in the PCV13 coverage required to demonstrate indirect effects of PCV13 qualitatively by site and in relation to vaccine schedule and use of catch-up campaigns. Inferential statistics are unlikely to be suitable with the inclusion of only three sites, and comparability between sites is limited due to variations between them.

### Power calculation

Power calculations were performed using nQuery Advisor+nTerim 4.0. Calculations were based on sample size methods for logistic regression models with a continuous covariate (ie, PCV coverage) and additional covariates, with inflation to account for clustering within villages. Power calculations assumed VT carriage prevalence of 30% in Lao PDR and 40% in Mongolia and PNG at the mean PCV coverage level[36] and VT carriage prevalence of 20% in Lao PDR and 30% in Mongolia and PNG at 1 SD above the mean PCV coverage level. Assuming a significance level of 0.05, allowing for adjustment using multiple covariates with an $R^2$ of 0.4, and that 50% of the cases are undervaccinated, a sample size of 1200 cases per site would provide between 87% and 92% power to determine the proportion of cases carrying VT pneumococcus at varying levels of village vaccine coverage. The power calculation has been adjusted to account for clustering by village, with higher variability in Lao PDR and PNG (intraclass coefficient (ICC) 0.1) and lower variability in Mongolia (ICC 0.01).

### Missing data

We will describe the number of participants with missing data on individual variables and compare the characteristics of those with and without missing data to determine whether there is evidence of systematic differences in characteristics. If we determine that the differences observed are able to be explained by available data (ie, missing at random), we will consider using multiple imputation to predict the distribution of the missing data in order to account for the bias due to incomplete data.[37]

### Patient and public involvement

Patients were not involved in the design of this study. Public health authorities in Laos and Mongolia were involved in the design and conduct of the study. In PNG, village representatives are approached to ensure that community surveys are conducted appropriately.

### Current status of the study

Recruitment is ongoing, and analysis of data, followed by publication of results, is expected from 2018 onwards. As of August 2017, we have recruited 1039, 481 and 3847 cases from Lao PDR, PNG and Mongolia, respectively. Table 4 describes the baseline characteristics of the children recruited.

### Ethics and dissemination

Prospective participants will be fully informed about the potential risks and benefits of participation, and written informed consent will be obtained prior to recruitment.

We plan on disseminating results to relevant stakeholders within Lao PDR, PNG and Mongolia, as well as submitting our findings for publication in relevant peer-reviewed journals and conferences.

## DISCUSSION

The ability for LMICto monitor the indirect effects of PCV is critical. Maximising indirect effects is important because these effects comprise a substantial proportion of overall PCV impact and increase the cost-effectiveness of the vaccine.[38] Indirect protection is particularly important for individuals who are unable to be vaccinated or who have poor vaccine responses, such as infants too young to be vaccinated and the elderly. In addition, the Bill & Melinda Gates Foundation is supporting studies looking into the effectiveness of reduced dose schedules (1+1). This reduced dose schedule needs to maintain indirect effects following PCV introduction, and this surveillance method provides a mechanism to determine when herd protection has been achieved and whether it is maintained.[39] Furthermore, the ability to determine whether substantial herd protection has been achieved may help to identify settings that are appropriate for introduction of reduced dose or modified schedules.

**Table 4** Case characteristics by site, Lao People's Democratic Republic (PDR), Mongolia and Papua New Guinea (PNG), 2014–2017

| | | Lao PDR n=1039, n (%) | Mongolia n=3847, n (%) | PNG n=481, n (%) |
|---|---|---|---|---|
| Year of recruitment | 2014 | 365/1039 (35) | NA | NA |
| | 2015 | 323/1039 (31) | 885/3847 (23) | NA |
| | 2016 | 281/1039 (27) | 2007/3847 (52) | 190/481 (40) |
| | 2017 | 70/1039 (7) | 955/3847 (25) | 291/481 (60) |
| Age group | <12 months | 432/1038 (42) | 1481/3847 (39) | 258/481 (54) |
| | 12–23 months | 346/1038 (33) | 1250/3847 (32) | 123/481 (25) |
| | ≥24 months | 260/1038 (25) | 1116/3847 (29) | 100/481 (21) |
| Gender | Male | 591/1039 (57) | 2079/3847 (54) | 278/481 (58) |

The results of our study will address the scarcity of literature determining the PCV coverage required to achieve substantial indirect effects, especially in LMIC. The inclusion of three sites, which have contrasting vaccine schedules, baseline intensities of pneumococcal carriage and healthcare systems, will also enable us to determine whether this vaccine coverage threshold differs by site and explore factors that may modify the vaccine coverage required to achieve indirect effects.

In this proposal, we describe a novel application of an analysis method to measure indirect effects. Although the isolation of a particular pneumococcal serotype from the nasopharynx of a child with ARI is not necessarily indicative of the serotype causing pneumococcal disease, reduced detection of VT pneumococci is likely to reflect reductions in disease due to VT pneumococci. Furthermore, our proposed analysis methods enable estimation of indirect effects in the absence of baseline pre-PCV data. This will be relevant for many LMIC that have little or no baseline data and are considering options for surveillance to accompany the introduction of PCV.

Another key strength of our methods is the use of consistent molecular serotyping microarray methods across all three sites, enabling sensitive detection of multiple serotype carriage and ascertainment of serotype-specific density.[28] Conventional pneumococcal carriage studies typically detect the presence of a single serotype and may overlook the presence of vaccine serotypes occurring at lower densities. The results of this study will also add to the limited literature on the effects of PCV on VT carriage density, which may affect likelihood of transmission.[40]

Our proposed methods have several potential limitations. First, there are significant challenges in determining accurate estimates of PCV coverage in resource-limited settings. In Mongolia and Lao PDR, where we are using administrative data, our estimates rely on the availability of accurate data about vaccine doses administered (numerator) as well as population estimates for the target age group (denominator). In both settings, we are conducting audits to assess the reliability of numerator data. Regarding denominator data, we are fortunate that a recent population census was conducted in Lao PDR in March 2015. However, we will be auditing denominator data in Mongolia, where population estimates are affected by large seasonal population movements between rural and urban settings. In PNG, the validity of the community surveys depends on representative sampling. However, survey participation can vary and is affected by season (related to farming practices), as well as community trust and understanding of the study. To maximise participation, we conducted mobile health clinics alongside the community surveys.

The second main limitation relates to detecting pneumococcal carriage in our study population of children hospitalised with respiratory infection. In this population, carriage detection may be affected by the prior use of antibiotics.[41 42] To address this, we have, where possible, aimed to recruit patients at admission, prior to receiving antibiotics or documented prior antibiotic use, which will be taken into account during analysis.

Third, there are variations in methods and setting across the three sites. This limitation may impact on the comparability of our results across the three sites. The reason for the differences in methods between sites is that our study is built on pre-existing studies (pneumonia aetiology studies in Lao PDR and PNG and a vaccine impact study in Mongolia) with established protocols.

To conclude, the results of this study will provide important feedback to national policy makers about the effects of newly introduced PCV programmes in Lao PDR, Mongolia and PNG. The results will help us understand the determinants of indirect effects and therefore guide strategies to maximise them. In particular, the results will also inform global policy about the vaccine coverage required to achieve substantial indirect effects in settings with different epidemiological characteristics of pneumococcal disease and carriage and will maximise the cost-effectiveness of the vaccine programmes.

**Author affiliations**
[1]Pneumococcal Research Group, Murdoch Children's Research Institute, Melbourne, Victoria, Australia
[2]Department of Paediatrics, The University of Melbourne, Melbourne, Victoria, Australia

[3]Global & Tropical Health Division, Menzies School of Health Research, Charles Darwin University, Darwin, Australia

[4]National Centre for Epidemiology & Population Health, Australian National University, Canberra, Australia

[5]School of Medicine, University of Western Australia, Perth, Australia

[6]Department of Infectious Diseases, Princess Margaret Hospital, Perth, Australia

[7]World Health Organization, Vientiane, Lao People's Democratic Republic

[8]Regional Office for the Western Pacific, World Health Organization, Manila, Philippines

[9]Infection and Immunity Unit, Papua New Guinea Institute of Medical Research, Goroka, Eastern Highlands, Papua New Guinea

[10]Institute for Infection and Immunity, St George's, University of London, London, UK

[11]BUGS Bioscience, London Bioscience Innovation Centre, London, UK

[12]Wesfarmers Centre for Vaccines and Infectious Diseases, Telethon Kids Institute, University of Western Australia, Perth, Australia

[13]National Center of Communicable Diseases (NCCD), Ministry of Health, Ulaanbaatar, Mongolia

[14]Centre for Tropical Medicine and Global Health, University of Oxford, Oxford, UK

[15]Lao-Oxford-Mahosot Hospital-Wellcome Trust Research Unit (LOMHWRU), Microbiology Laboratory, Mahosot Hospital, Vientiane, Lao People's Democratic Republic

[16]National Immunization Programme, Ministry of Health, Vientiane, Lao People's Democratic Republic

[17]Department of Microbiology and Immunology, The University of Melbourne at the Peter Doherty Institute for Infection and Immunity, Melbourne, Victoria, Australia

[18]Department of Infectious Disease Epidemiology, London School of Hygiene & Tropical Medicine, London, UK

[19]Centre for International Child Health, Department of Paediatrics, The University of Melbourne, Melbourne, Victoria, Australia

**Acknowledgements** We would like to acknowledge the Ministries of Health of Lao PDR and Mongolia, LOMHWRU, PNG IMR and WHO and support from the Victorian Government's Operational Infrastructure Support Program. We would also like to thank study staff, laboratory staff and participating families.

**Collaborators** The PneuCAPTIVE protocol development group includes the authors of the paper listed in the byline and the following: Dashtseren Luvsantseren (NCCD, Ulaanbaatar, Mongolia), Bujinlkham Suuri (NCCD, Ulaanbaatar, Mongolia), Mukhchuluun Ulziibayar (NCCD, Ulaanbaatar, Mongolia), Dashpagam Otgonbayer (NCCD, Ulaanbaatar, Mongolia), Audrey Dubot-Pérès (LOMHWRU, Vientiane, Lao PDR), Keodomphone Vilavong (LOMHWRU, Vientiane, Lao PDR), Anisone Chanthongthip (LOMHWRU, Vientiane, Lao PDR), Syladeth Chanthaphone (LOMHWRU, Vientiane, Lao PDR), Joycelyn Sapura (PNG IMR, Goroka, PNG), John Kave (PNG IMR, Goroka, PNG), Tonny Kumani (PNG IMR, Goroka, PNG) and Wendy Kirarock (PNG IMR, Goroka, PNG).

**Contributors** FMR conceived the idea and designed the study. JYRL, SD, KF, PNN, RL, RP, AX, DABD and FMR supported the development of country-specific protocols and study implementation in Lao PDR. CCB, RF, DL, WP and FMR supported the development of country-specific protocols and study implementation in PNG. TM, SLV, CvM, KM, JC and FR supported the development of country-specific protocols and study implementation in Mongolia. CS, EMD and JH devised the microbiological approach and laboratory protocols. JC, CDN, RA and FR devised the analysis plan. JC and FR drafted the manuscript. All authors provided feedback to the draft manuscript and have read and approved the final version.

**Funding** This work is supported by the Bill & Melinda Gates Foundation grant number (OPP1115490). JC is completing a PhD at The University of Melbourne, funded by an Australian Government Research Training Program scholarship.

**Competing interests** None declared.

**Patient consent** Guardian consent obtained.

**Ethics approval** The study is being conducted according to protocols approved by the following ethics committees: Lao PDR Ministry of Health National Ethics Committee for Health Research (057/2013 NECHR), Oxford Tropical Research Ethics Committee (1050-13), Mongolian National Ethics Committee for Health Research, the WHO Regional Office for the Western Pacific (WPRO) Ethics Review Committee (2013.30.LAO.2.EPI, Mongolia), PNG IMR Institutional Review Board (1510), Government of PNG Medical Research Advisory Committee (15.18) and the Royal Children's Hospital/MCRI Human Research Ethics Committee (33177B and 33203E).

**Provenance and peer review** Not commissioned; externally peer reviewed.

**Data sharing statement** We have an agreement with Bill & Melinda Gates Foundation to share datasets that are requested by non-profit institutions and/or scientific researchers for a particular purpose, such as a meta-analysis or systematic review. We would make them available after ensuring appropriate ethical considerations. Proposals should be directed to Associate Professor Fiona Russell (fmruss@unimelb.edu.au).

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
