## [Reviewer comments · BMJ Open]

ARTICLE DETAILS

TITLE (PROVISIONAL)	Determining the pneumococcal conjugate vaccine coverage required for indirect protection against vaccine-type pneumococcal carriage in low- and middle-income countries: a protocol for a prospective observational study
AUTHORS	Chan, Jocelyn; Nguyen, Cattram; Lai, Jana; Dunne, Eileen; Andrews, Ross; Blyth, Christopher C.; Datta, Siddhartha; Fox, Kim; Ford, Rebecca; Hinds, Jason; La Vincente, Sophie; Lehmann, Deborah; Lim, Ruth; Mungun, Tuyu; Newton, Paul; Phetsouvanh, Rattanaphone; Pomat, Willie; Xeuatvongsa, Anonh; von Mollendorf, Claire; Dance, David; Satzke, Catherine; Muholland, Kim; Russell, F. M.

VERSION 1 – REVIEW

REVIEWER	Anneke Steens Norwegian Institute of Public Health, Norway
REVIEW RETURNED	22-Jan-2018

GENERAL COMMENTS	Review of "Determining the pneumococcal conjugate vaccine coverage required for indirect immunity in low- and middle-income countries: a protocol for a prospective observational study" written by Chan et al. General comments: The manuscript described a protocol for a study that will determine the vaccine coverage threshold to achieve indirect protection of PCV on carriage by ARI patients (as indication of the indirect effect on disease). As the study states, there is a gap in such knowledge, particularly from low- and middle-income countries. They use a promising design and compare different countries, which is promising for interesting results. I have some commends for clarification. Specific comments: Title • In the title you write "indirect immunity". I think this should be "indirect protection" (PCV does not provide exposure of non-vaccinated individuals like e.g. oral polio vaccine does). Abstract: • Page 4, line 21: define a case (combined in the first sentence of the method?). • Page 4, line 33: change "maximise the effects of PCV" with
---

“achieve indirect protection” (as that is your aim, isn't it?).

Introduction:

- Page 5, line 36-37: define indirect effects on what: carriage, respiratory infections, ARI and/or IPD?
- Page 6, line 2-6: in an Australian study they concluded that the booster seemed particularly important for obtaining indirect protection because of their lower indirect effects than in other western setting (reference Jayasinghe S. Clin Infect Dis 2017; 64: 175–83.)
- Page 6, line 22-23: maybe change “as part of the PneuCAPTIVE” to “in the study called PneuCAPTIVE” (to me as a reader it is not clear what PneuCAPTIVE otherwise includes)

Objectives

- Page 6, line 54: define a case (maybe already in the text above) and its contact

Methods:

- Page 7, line 31: add . and remove the paragraph on PNG from “(The PNG PneumoCAPTIVE...”,
- Page 7, line 24: place a reference to Table 1
- Page 7, line 33: what does MCRI stand for?
- Page 7, line 48: remove “Table 1)”
- Page 9, Table 2: can you add information on the study population (i.e., how many children are there in the catchment area of the included hospitals and how representative will your cases be for the population (thinking of e.g. urban / rural)? Furthermore, can you add information on the study period (data sampling period) in the different sites?
- Page 9, Table 2: write in the 3rd row: “definition of ARI”. In the 3rd row under Lao PDR: what do you mean with “history of fever” (besides measured), is it reported by the parent?
- Page 9, Table 2: for sampling in Mongolia: maybe change to “randomly selected from all children coming to the hospital” (is that correct?)
- Page 9, line 55: are samples of all 3 countries send to MCRI?
- Page 10, line 8-9: it seems that there are missing some words in this sentence.
- Page 10, line 23: how will you classify children carrying both VT and non-VT pneumococci?
- Page 10, line 31: will this definition also be used for the controls sampled in PNG? Maybe then just write “vaccine history will be defined as...”
- Page 10, line 44: Can you provide more information about the surveys that you do in PNG? How many children do the surveys cover? What is the sampling procedure? Do you use the optimum-sized neighbourhood method as described in the reference Khatib 2012 Lancet infectious diseases? And is there information about the reliability of the administrative data in Lao PDR? How do you select the administrative records for a village or subdistrict – can you select on postal code?
- Page 10, line 44: Do you determine the vaccine coverage in villages from which cases arose both for the VT and non-VT ARI cases? That seems essential. Please specify.
- Page 11, line 31: do you mean the presence of VT carriage per child (instead of the prevalence of VT carriage)? I suppose this is a yes/no variable?
- Page 12, line 5-8: there are inconclusive results on whether or not replacement carriage is complete after PCV13 (see e.g. Steens

	2015 and Ricketson LJ 2014. Furthermore, Steens 2016 in PeerJ showed that the carriage 4 years after vaccine introduction in Norway was still only 50% while this was 80% before PCV introduction). This methods therefore seems “risky”, though, your descriptive analyses will show whether or not overall carriage remains at pre-vaccination levels.  • Page 12, line 13: what do you mean with rolling 7-months intervals? Do you mean you will smooth over a 7-month period? (I now see you write that in the line 17-18 -> combine to one sentence?) • Page 12, line 15: about age-differences: to me it seems most correct not to include care-givers in the analysis of the controls, but only community controls in the same age, because of the strong association between age and carriage and no overlap in (older) age-distribution. For objective 3 it seems useful to me to include also the care-givers. Can you clarify what you are planning to do for which objective? • Page 12, line 27: see comment on Page 11, line 31: do you mean the presence of VT carriage per child? Or specify what kind of model you will use. • Page 12, line 46: you assumed 20% carriage in Lao PDR and Mongolia. Do you have data on these countries from other studies? The prevalence seems very low for the age group. A prevalence closer to 50% will likely increase your sample size, so the sample size seems maybe too small in these locations (though, you have very high power, so it might not be a problem). Discussion:  • Page 14, line 22 and 23: change herd immunity to herd protection (see comment on title) • Page 14, line 35: it is not a novel surveillance method (as you also refer to the Cholera study), though, it might be a novel application of the method (namely to indirect effects)? Maybe change the wording? • Page 14, line 51: change a lower densities in at lower densities.
--	--

REVIEWER	ANGEL VILA-CORCOLES iNSTITUT cATALÀ DE LA SALUT (ICS) SPAIN
REVIEW RETURNED	23-Jan-2018

GENERAL COMMENTS	GENERAL COMMENT: This is a manuscript describing the study protocol of an ongoing multicenter observational study aimed to investigate pneumococcal nasopharyngeal carriage (PNPC) among children 2-5 years in 3 Asia-Pacific settings (Lao, Mongolia and Papua New Guinea), to assess relationships between prevalence of NPC and PCV13 coverages and to determine which would be the minimum PCV13 coverage required to expect an indirect (herd) effect among unvaccinated people. The study question is important because, while indirect effects of PCV vaccination have been described, the PCV coverage required to achieve is unknown. The paper is well written and it is scientifically accurate. References are appropriate and updated, and tables contain relevant information in order to better understand methodology and interpret future results. I have few comments.
---

	1.- Abstract. The objective in the abstract is few descriptive and it may be confusing (“We will investigate “this” using hospital-based NP pneumococcal carriage surveillance at three sites in the Asia-Pacific region”). This phrase describes methods and settings but it does not contain a clear objective. I suggest to rewrite it according to the specific objectives of the study mentioned in the full text of the manuscript. 2.- Strengths and Limitations (below the Abstract) The reduction of nasopharyngeal carriage for some specific-pneumococcal serotypes among vaccinated and undervaccinated children with acute respiratory infections do not necessarily imply an effective indirect effect in preventing pneumococcal disease in the population. This should be commented as limitation here. 3.- Methods. Does the “Undervaccinated” group include never vaccinated? It seems that the definition for “undervaccinated” includes as undervaccinated (few/indadequate number of PCV13 doses) as well as never vaccinated. Logically, immunity in both cases would be distinct and, so, it would be clarified and/or commented in the Discussion/limitations. 4.- Considering that “nasopharyngeal carriage” is the key point in this study, I think that this term should be included in the title.
--	---

VERSION 1 – AUTHOR RESPONSE

Reviewers' Comments to Author:

Reviewer: 1

Reviewer Name: Anneke Steens

Institution and Country: Norwegian Institute of Public Health, Norway Competing Interests: None declared

General comments:

The manuscript described a protocol for a study that will determine the vaccine coverage threshold to achieve indirect protection of PCV on carriage by ARI patients (as indication of the indirect effect on disease). As the study states, there is a gap in such knowledge, particularly from low- and middle-income countries. They use a promising design and compare different countries, which is promising for interesting results.

I have some commends for clarification.

Specific comments:

Title

- In the title you write “indirect immunity”. I think this should be “indirect protection” (PCV does not provide exposure of non-vaccinated individuals like e.g. oral polio vaccine does).

Authors' response: Thank you – we have changed “indirect immunity” to “indirect protection” in the title.

Abstract:

- Page 4, line 21: define a case (combined in the first sentence of the method?).

Authors' response: We have amended the first sentence of the methods to define cases as follows: "We are recruiting cases, defined as children aged 2-59 months admitted to participating hospitals with acute respiratory infection in Lao People's Democratic Republic, Mongolia and Papua New Guinea."

- Page 4, line 33: change "maximise the effects of PCV" with "achieve indirect protection" (as that is your aim, isn't it?).

Authors' response: We have changed to "achieve indirect protection" in the final sentence of the abstract.

Introduction:

- Page 5, line 36-37: define indirect effects on what: carriage, respiratory infections, ARI and/or IPD?

Authors' response: We have changed the sentence to: "The threshold vaccination coverage required to achieve significant indirect PCV effects on either pneumococcal carriage or disease outcomes are not well understood."

- Page 6, line 2-6: in an Australian study they concluded that the booster seemed particularly important for obtaining indirect protection because of their lower indirect effects than in other western setting (reference Jayasinghe S. Clin Infect Dis 2017; 64: 175–83.)

Authors' response: Thank you for highlighting - we have added this sentence: "A recent study from Australia, the only high-income country to use the 3+0 schedule, concluded that the booster dose may be important for obtaining indirect protection because of the lower indirect effects observed compared to other high-income country settings."

- Page 6, line 22-23: maybe change "as part of the PneuCAPTIVE" to "in the study called PneuCAPTIVE" (to me as a reader it is not clear what PneuCAPTIVE otherwise includes)

Authors' response: We have changed "as part of the PneuCAPTIVE" to "in the study called PneuCAPTIVE".

Objectives

- Page 6, line 54: define a case (maybe already in the text above) and its contact

Authors' response: We have amended the paragraph to define cases and contacts: (1) investigate the relationship between PCV13 coverage and VT carriage among under-vaccinated cases, defined as children (indirect effects) aged 2-59 months with an ARI in ...; (2) describe monthly trends in VT carriage prevalence among cases and contacts. Contacts are defined as children 0-59 months of age, who have slept in the same house as or played with the case during the preceding three weeks;

Methods:

- Page 7, line 31: add . and remove the paragraph on PNG from "(The PNG PneumoCAPTIVE...)",

Authors' response: Done. (Note that this duplicated paragraph was not in the original word document, but may have been introduced in the upload process, so there are no tracked changes).

- Page 7, line 24: place a reference to Table 1

Authors' response: Added, see page 6, line 24.

- Page 7, line 33: what does MCRI stand for?

Authors' response: We have defined MCRI (Murdoch Children's Research Institute) on page 6, line 29.

- Page 7, line 48: remove "Table 1"

Author's response: This has now been removed.

- Page 9, Table 2: can you add information on the study population (i.e., how many children are there in the catchment area of the included hospitals and how representative will your cases be for the population (thinking of e.g. urban / rural)? Furthermore, can you add information on the study period (data sampling period) in the different sites?

Authors' response: We have added the following information on sampling period and study population to Table 2.

Site	Lao PDR	Mongolia	PNG
...			
Sampling period	December 2013–November 2019 April 2016 –March 2019	December 2013–November 2019	November 2015 –October 2018†
Study population	Mahosot Hospital is one of two tertiary-level paediatric hospital in Vientiane and receives a mix of patients from urban Vientiane city, rural Vientiane province and other provinces. The two secondary-level district hospitals and a tertiary-level Maternal and Child Health hospital service the vast majority of children in the two districts that received PCV. There are a limited number of paediatric beds at private hospitals in Ulaanbaatar. The Eastern Highlands Provincial hospital is the sole hospital for the province. Study population includes urban and rural households within 1 hour drive of Goroka.		

† A one-year extension (until June 2019) has been sought for the Mongolian site

- Page 9, Table 2: write in the 3rd row: "definition of ARI". In the 3rd row under Lao PDR: what do you mean with "history of fever" (besides measured), is it reported by the parent?

Authors' response: We have added a title for the 3rd row ("Definition of acute respiratory infection") and changed fever "history" to "parent report" as below.

Site	Lao PDR	Mongolia	PNG
...			
Definition of acute respiratory infection	Fever (parent report or measured) AND one of cough OR dyspnoea OR rhinitis OR abnormal chest auscultation Cough OR dyspnoea AND tachypnoea* OR hypoxia OR chest indrawing Cough AND tachypnoea* AND lower chest wall indrawing		

- Page 9, Table 2: for sampling in Mongolia: maybe change to "randomly selected from all children coming to the hospital" (is that correct?)

Authors' response: We have changed Table 2 for sampling in Mongolia to "A random sample (33 per month) of all enrolled cases are selected for testing"

- Page 9, line 55: are samples of all 3 countries send to MCRI?

Authors' response: Yes, we have amended the sentence to specify: "Swabs are vortexed... and transported from all three sites to the Pneumococcal Research Laboratory at MCRI"

- Page 10, line 8-9: it seems that there are missing some words in this sentence.

Authors' response: We have deleted an additional "the".

- Page 10, line 23: how will you classify children carrying both VT and non-VT pneumococci?

Authors' response: We have clarified by adding this sentence: "In the context of multiple serotype carriage, VT carriage will be defined as the presence of at least one VT regardless of the presence of other serotypes."

- Page 10, line 31: will this definition also be used for the controls sampled in PNG? Maybe then just write "vaccine history will be defined as..."

Authors' response: We have reworded as suggested.

- Page 10, line 44: Can you provide more information about the surveys that you do in PNG? How many children do the surveys cover? What is the sampling procedure? Do you use the optimum-sized neighbourhood method as described in the reference Khatib 2012 Lancet infectious diseases? And is there information about the reliability of the administrative data in Lao PDR? How do you select the administrative records for a village or subdistrict – can you select on postal code?

Authors' response: We have added this sentence to describe sampling procedure and neighbourhood boundaries for the surveys: "We are surveying all children less than five years of age, from households within 5 minutes' walk of the case, since this is the group of children with whom the case is mostly likely to interact and therefore influence their carriage status". Unfortunately, due to the lack of household-level GIS data, we were not able to apply the method described in the reference Khatib 2012 Lancet ID to specify optimum neighbourhood size.

We have also added the following to the manuscript: "To determine the reliability of our methods in Lao PDR, we plan on comparing our coverage estimates with a National Immunisation Survey, conducted according to WHO guidelines in 2015, when provincial level estimates from this survey become available.

In Mongolia, each subdistrict is numbered, allowing for easy identification of relevant subdistricts for each child. In Lao PDR, parents are often unaware of the administrative code for their resident villages, therefore the allocation of an administrative village code for each child is performed by local staff based on the full address (province, district, village). We have added a sentence to specify that: "The resident village or subdistrict is identified using relevant administrative codes, which are determined by local staff on enrolment. This determines the health centre's administrative boundary for the provision of immunisation services"

- Page 10, line 44: Do you determine the vaccine coverage in villages from which cases arose both for the VT and non-VT ARI cases? That seems essential. Please specify.

Authors' response: Yes we are determining vaccine coverage for both VT and non-VT ARI cases. We have amended the sentence: "For each under-vaccinated case recruited (including those with and without VT pneumococci), we are..."

- Page 11, line 31: do you mean the presence of VT carriage per child (instead of the prevalence of VT carriage)? I suppose this is a yes/no variable?

Authors' response: This is a yes/no variable, so we have deleted the word "prevalence" where appropriate throughout the manuscript.

- Page 12, line 5-8: there are inconclusive results on whether or not replacement carriage is complete after PCV13 (see e.g. Steens 2015 and Ricketson LJ 2014. Furthermore, Steens 2016 in PeerJ showed that the carriage 4 years after vaccine introduction in Norway was still only 50% while this was 80% before PCV introduction). This methods therefore seems "risky", though, your descriptive analyses will show whether or not overall carriage remains at pre-vaccination levels.

Authors' response: We agree there is uncertainty regarding the time period required for replacement carriage to be complete. If descriptive analyses show that overall carriage have not returned to pre-vaccination levels, we may need to consider extending our study. At our first site, in Laos, we have recently been awarded an extension to continue data collection for five years post-PCV introduction. We are also seeking an extension in Mongolia. We have added text and footnotes to table 2. In PNG, interim results from 2016, 3 years post PCV introduction, indicate replacement carriage is complete.

- Page 12, line 13: what do you mean with rolling 7-months intervals? Do you mean you will smooth over a 7-month period? (I now see you write that in the line 17-18 -> combine to one sentence?)

Authors' response: We have combined the two sentences into the following one: "Crude and adjusted monthly VT carriage prevalence will be assessed within rolling seven-month intervals to present smooth curves and assess trends over time"

- Page 12, line 15: about age-differences: to me it seems most correct not to include care-givers in the analysis of the controls, but only community controls in the same age, because of the strong association between age and carriage and no overlap in (older) age-distribution. For objective 3 it seems useful to me to include also the care-givers. Can you clarify what you are planning to do for which objective?

Authors' response: Yes we agree, the analysis for Objective 2 will only include contacts, whereas the analysis for Objective 3 will be conducted for both caregivers and contacts. We have also amended the objectives to reflect this:

"(2) describe monthly trends in VT carriage prevalence among cases and contacts. Contacts are defined as children 0-59 months of age, who have slept in the same house as or played with the case during the preceding three weeks; (3) investigate the relationship between PCV13 coverage and VT carriage among under-vaccinated contacts (indirect effects) and caregivers living in the community"

- Page 12, line 27: see comment on Page 11, line 31: do you mean the presence of VT carriage per child? Or specify what kind of model you will use.

Authors' response: As above, we have changed "prevalence" to "presence" of VT carriage per child.

- Page 12, line 46: you assumed 20% carriage in Lao PDR and Mongolia. Do you have data on these countries from other studies? The prevalence seems very low for the age group. A prevalence closer to 50% will likely increase your sample size, so the sample size seems maybe too small in these locations (though, you have very high power, so it might not be a problem).

Author's response: Thank you for highlighting this. Our initial sample size calculations were based on interim data from the three sites. In Lao PDR, overall pneumococcal carriage was 43.2% and VT carriage was 22.4% for cases enrolled 2013-2014 (lower PCV coverage). In Mongolia, overall pneumococcal carriage was 48.5% and VT carriage was 34.5% for cases enrolled in 2016 (pre-PCV). In PNG, overall pneumococcal carriage was 88% and VT carriage was 35.8% for cases enrolled in 2016 (low PCV coverage). As such, we have increased the assumed VT carriage rates to 30% in Lao PDR and 40% in Mongolia to avoid under-estimating sample size (or over-estimating power). The revised power calculations indicate: "a sample size of 1200 cases per site would provide between 87 and 92% power."

Discussion:

- Page 14, line 22 and 23: change herd immunity to herd protection (see comment on title)

Authors' response: We have changed "herd immunity" to "herd protection".

- Page 14, line 35: it is not a novel surveillance method (as you also refer to the Cholera study), though, it might be a novel application of the method (namely to indirect effects)? Maybe change the wording?

Authors' response: We have changed the wording to: "we describe a novel application of an analysis method to measure indirect effects"

- Page 14, line 51: change a lower densities in at lower densities.

Authors' response: We have changed "a" to "at".

Reviewer: 2

Reviewer Name: ANGEL VILA-CORCOLES

Institution and Country: INSTITUT CATALÀ DE LA SALUT (ICS), SPAIN Competing Interests: NONE DECLARED

GENERAL COMMENT:

This is a manuscript describing the study protocol of an ongoing multicenter observational study aimed to investigate pneumococcal nasopharyngeal carriage (PNPC) among children 2-5 years in 3 Asia-Pacific settings (Lao, Mongolia and Papua New Guinea), to assess relationships between prevalence of NPC and PCV13 coverages and to determine which would be the minimum PCV13 coverage required to expect an indirect (herd) effect among unvaccinated people.

The study question is important because, while indirect effects of PCV vaccination have been described, the PCV coverage required to achieve is unknown.

The paper is well written and it is scientifically accurate. References are appropriate and updated, and tables contain relevant information in order to better understand methodology and interpret future results.

I have few comments.

1.- Abstract. The objective in the abstract is few descriptive and it may be confusing ("We will investigate "this" using hospital-based NP pneumococcal carriage surveillance at three sites in the Asia-Pacific region"). This phrase describes methods and settings but it does not contain a clear objective. I suggest to rewrite it according to the specific objectives of the study mentioned in the full text of the manuscript.

Authors' response: We have amended the sentence to: "We will investigate the relationship between PCV13 coverage and VT carriage among under-vaccinated children using hospital-based NP pneumococcal carriage surveillance at three sites in the Asia-Pacific region."

2.- Strengths and Limitations (below the Abstract) The reduction of nasopharyngeal carriage for some specific-pneumococcal serotypes among vaccinated and undervaccinated children with acute respiratory infections do not necessarily imply an effective indirect effect in preventing pneumococcal disease in the population. This should be commented as limitation here.

Authors' response: We have added this limitation to the manuscript: "This method does not measure indirect protection against disease. However, a reduction in VT carriage among undervaccinated cases indicates likely reduction in VT disease, since carriage is a precursor for disease."

3.- Methods. Does the "Undervaccinated" group include never vaccinated?

It seems that the definition for "undervaccinated" includes as undervaccinated (few/indadequate number of PCV13 doses) as well as never vaccinated. Logically, immunity in both cases would be distinct and, so, it would be clarified and/or commented in the Discussion/limitations.

Authors' response: Yes, the under-vaccinated group includes never vaccinated. It is true the immunity between these groups may be distinct. We have added in an additional sensitivity analysis: "To determine whether a higher PCV coverage is required to achieve indirect effects among completely un-vaccinated cases compared to under-vaccinated cases, we will conduct a sensitivity analysis among children who have never received PCV."

4.- Considering that "nasopharyngeal carriage" is the key point in this study, I think that this term should be included in the title.

Authors' response: We have amended the title of the manuscript to: "Determining the pneumococcal conjugate vaccine coverage required for indirect protection against vaccine-type pneumococcal carriage in low- and middle-income countries: a protocol for a prospective observational study"

VERSION 2 – REVIEW

REVIEWER	Angel Vila-Córcoles INSTITUT CATALÀ DE LA SALUT. Spain
REVIEW RETURNED	27-Feb-2018

GENERAL COMMENTS	No comments
-------------

REVIEWER	Anneke Steens Norwegian Institute of Public Health Norway
REVIEW RETURNED	13-Mar-2018

GENERAL COMMENTS	 • Page 4, line 28: higher rate of coverage: do you mean vaccine coverage (then just write higher vaccine coverage), or do you mean the coverage/percentage of IPD cases covered by VT serotypes? • Page 4, line 49: change "maximise the indirect effects" to "achieve indirect protection" (as that is your aim, isn't it?). • Page 5, line 29 (as well as page 3, line 43): I think that the statement that a reduction in VT carriage in ARI cases reflects a reduction in VT disease is too strong. You will not know whether ARI is caused by the pneumococci. So, I do agree that you would expect that disease has gone down is carriage has gone down, but the size
---

	of the reduction does not need to be the same. Maybe use a less strong word than “reflect”, as that indicates that the size would be similar?  • Page 6, line 10: In page 5, line 19-20 it says that baseline data is not available, or do I misunderstand it? • Page 6, line 38: add: sampling started in December 2013. • Page 6, line 51: add: sampling started in November 2015. • Page 7, line 7: add: sampling started in April 2016. • Page 9, line 37: Will children with multiple carriage count twice, i.e., both as a VT and NVT case in case they carry both? If they would only be counted as VT case, you may introduce bias. • Page 10, line 12: by only including the number of children that live within 5 minutes walk from the case, the village surveys will be underpowered to determine the vaccine coverage in the area, I assume. As you want to look at indirect effects, you should cover the area “which is relevant for transmission in the area (not only directly to the case)”, which is likely larger than what you are including now. I understand it is a lot of work to include more children, but I question whether you can trust the data you will get with such strategy. Try to increase the sample size per village survey. If that is not feasible, try at least to do that in a subsample and compare in the subsamples the coverage you would get by only including those living within 5 minutes, and living in the larger area, as a sensitivity analysis. • Page 12, line 38: if there is systemic differences, you cannot use imputation to account for bias. In the paper you refer to I read: “Unfortunately, it is not possible to distinguish between missing at random and missing not at random using observed data. Therefore, biases caused by data that are missing not at random can be addressed only by sensitivity analyses examining the effect of different assumptions about the missing data mechanism..... In such cases multiple imputation may give misleading results.” • Page 12, line 43: patients were not involved in what?
--	---

VERSION 2 – AUTHOR RESPONSE

Editorial Request:

Can you please revise the first bullet point of the strengths and limitations on page 3? This appears to be a general summary rather than stating a specific strength of the study relating to its design/ methods.

Authors’ response: We have amended the bullet point to emphasize the strengths of the methods: “We describe a novel application of study methods that enable monitoring the indirect effects of pneumococcal conjugate vaccines (PVCs) using pneumococcal carriage and in the absence of baseline data”

Reviewers’ Comments to Author:

Reviewer: 1

Reviewer Name: Anneke Steens

Institution and Country: Norwegian Institute of Public Health, Norway Competing Interests: None declared

- Page 4, line 28: higher rate of coverage: do you mean vaccine coverage (then just write higher vaccine coverage), or do you mean the coverage/percentage of IPD cases covered by VT serotypes?

Authors' response: We have changed the phrase to "higher vaccine coverage" rather than "higher rate of coverage".

- Page 4, line 49: change "maximise the indirect effects" to "achieve indirect protection" (as that is your aim, isn't it?).

Authors' response: We have changed to phrase to "achieve indirect protection".

- Page 5, line 29 (as well as page 3, line 43): I think that the statement that a reduction in VT carriage in ARI cases reflects a reduction in VT disease is too strong. You will not know whether ARI is caused by the pneumococci. So, I do agree that you would expect that disease has gone down is carriage has gone down, but the size of the reduction does not need to be the same. Maybe use a less strong word than "reflect", as that indicates that the size would be similar?

Authors' response: We have changed the wording from "reflects" to "suggests" at both points in the manuscript, to communicate the uncertainty regarding the size of reduction. The full sentence reads as follows: "A reduction in vaccine-type (VT) pneumococcal carriage suggests likely reductions in disease due to VT pneumococci, since carriage is a precursor to disease."

- Page 6, line 10: In page 5, line 19-20 it says that baseline data is not available, or do I misunderstand it?

Authors' response: We have removed the word "baseline" on page 6, line 10. The sentence reads: ... and determine the degree to which site-specific factors, such as pneumococcal carriage rates and densities, vaccine schedule and use of catch-up campaigns, account for differences in the PCV13 coverage required to demonstrate indirect effects.

- Page 6, line 38: add: sampling started in December 2013.

Authors' response: Added

- Page 6, line 51: add: sampling started in November 2015.

Authors' response: Added

- Page 7, line 7: add: sampling started in April 2016.

Authors' response: Added

- Page 9, line 37: Will children with multiple carriage count twice, i.e., both as a VT and NVT case in case they carry both? If they would only be counted as VT case, you may introduce bias.

Authors' response: Yes, that is correct – the children will be counted twice.

- Page 10, line 12: by only including the number of children that live within 5 minutes walk from the case, the village surveys will be underpowered to determine the vaccine coverage in the area, I assume. As you want to look at indirect effects, you should cover the area "which is relevant for transmission in the area (not only directly to the case)", which is likely larger than what you are including now. I understand it is a lot of work to include more children, but I question whether you can trust the data you will get with such strategy. Try to increase the sample size per village survey. If that is not feasible, try at least to do that in a subsample and compare in the subsamples the coverage

you would get by only including those living within 5 minutes, and living in the larger area, as a sensitivity analysis.

Authors' response: We appreciate your suggestions and understand your concerns about sample size. In PNG, the study team are aware of the need to cover the area relevant for transmission. On review with the field team, surveys commonly include children within a 10 minute walk, but not more than 20 minutes' walk (we have amended this in the protocol). Our methods are largely dictated by the geography of the region. The majority of cases live in rural areas comprised of discrete villages, with children predominantly interacting with other children within the same village.

- Page 12, line 38: if there is systemic differences, you cannot use imputation to account for bias. In the paper you refer to I read: "Unfortunately, it is not possible to distinguish between missing at random and missing not at random using observed data. Therefore, biases caused by data that are missing not at random can be addressed only by sensitivity analyses examining the effect of different assumptions about the missing data mechanism..... In such cases multiple imputation may give misleading results."

Authors' response: We have amended the missing data section to clarify the circumstances under which we would consider multiple imputation. "If we determine that the differences observed are able to be explained by available data (i.e. missing at random), we will consider using multiple imputation to account for the bias due to incomplete data."

For your convenience I have included a few key definitions from the paper quoted above for reference.

Box 1 Types of missing data – from Sterne et al. BMJ. 2009.

- o Missing completely at random—There are no systematic differences between the missing values and the observed values. For example, blood pressure measurements may be missing because of breakdown of an automatic sphygmomanometer
- o Missing at random—Any systematic difference between the missing values and the observed values can be explained by differences in observed data. For example, missing blood pressure measurements may be lower than measured blood pressures but only because younger people may be more likely to have missing blood pressure measurements
- o Missing not at random—Even after the observed data are taken into account, systematic differences remain between the missing values and the observed values. For example, people with high blood pressure may be more likely to miss clinic appointments because they have headaches

- Page 12, line 43: patients were not involved in what?

Authors' response: We have changed the sentence to specify: "Patients were not involved in the design of this study."

Reviewer: 2

Reviewer Name: Angel Vila-Córcoles

Institution and Country: INSTITUT CATALÀ DE LA SALUT. Spain Competing Interests: None declared

No comments

VERSION 3 – REVIEW

REVIEWER	Annek Steens Norwegian Institute of Public Health
REVIEW RETURNED	03-Apr-2018
GENERAL COMMENTS	No more comments. Good Luck with the continuation of the study.